# OpenReview forum: "The Athenian Academy: A Seven-Layer Architecture for Multi-Agent Systems"
_ICLR.cc/2026/Conference — Submitted to ICLR 2026_

### Official Review · Reviewer_iEEj · 2025-10-27

**Soundness:** 1
**Presentation:** 1
**Contribution:** 1
**Rating:** 0
**Confidence:** 4

**Summary:**

This work proposes a seven-layer architecture for a structured MAS design. The architecture progresses from the micro-level agent design to the macro-level design. Precisely, it includes inter-agent dynamics (layer 1), intra-agent adaptability (layers 2-4), agent-model interaction patterns (layers 5-6), and an agent-system synthesis (layer 7) layer. It conducts human evaluation to validate the design of each layer.

**Strengths:**

The proposed framework provides a conceptual framework for designing current multi-agent systems.

**Weaknesses:**

1. There is no technical contribution given in this work, but rather a conceptual framework for guiding multi-agent design. All layers have been widely discussed by the community. Consequently, the proposed layer also lacks significant novelty.

2. Secondly, the layers defined in this work do not clearly differentiate one from another. For example, both single-agent multi-role playing (layer 2) and single-agent multi-capability avatars (layer 4) indicate a similar concept.

3. The experiments conducted in this work lack soundness. The evaluations for each layer were only conducted by a very small group of human evaluators. For instance, layer 1 validation includes only 2 graduate students for a Likert scale evaluation. The experiments were also not conducted on well-established benchmarks or datasets. Therefore, it is hard to justify the gain of any performance improvements.

4. Lastly, the empirical results lack reproducibility. This work does not include any experimental scripts, the detailed prompts, and generated artifacts per the validation of each layer.

**Questions:**

Could you clarify the experimental details, such as the original prompts for conducting each experiment?

**Details Of Ethics Concerns:**

Figure 1 includes Raphael’s The School of Athens.

---

### Official Review · Reviewer_mYxo · 2025-11-01

**Soundness:** 2
**Presentation:** 2
**Contribution:** 1
**Rating:** 2
**Confidence:** 4

**Summary:**

The paper proposes The Athenian Academy, a seven-layer architecture for LLM-based multi-agent systems. It offers a structured ethodology for designing, implementing, and evaluating complex agent systems.

**Strengths:**

1. This paper identifies an important problem in multi-agent systems. It is significant to conduct research on unifying MAS architecture to improve generality.
2. The paper is well-structured, and it clearly conveys the seven layers of the proposed MAS architecture.
3. The selected experiments for each layer are concrete and precise.

**Weaknesses:**

1. This paper states that it addresses the fragmented and ad-hoc MAS design. However, MAS covers most of the complex systems (e.g., society, industry, network, etc.), and agents could be physical or functional. The architecture is not justified or proven to be an efficient way for all settings.
2. The seven layers are all common-sense perspectives of MAS. Though the layered hierarchy is initially proposed by this paper, it is hard to see the relationships between layers and why this hierarchy makes sense.
3. The agents in this paper seem to be just LLM models without any agent design. The single experiments in art creation cannot support the contributions that the paper states. inclu
4. The validations in each layer are pretty ad-hoc as well. The experiments are not well-validated (e.g., enough human evaluations) and did not compare benchmarks.

**Questions:**

1. What are the relationships between the layers?
2. What are the new concepts or designs compared to existing works in each layer?
3. How are the metrics chosen in each layer?
4. How can people use this architecture for their own MAS scenario?

---

### Official Review · Reviewer_akP7 · 2025-11-01

**Soundness:** 2
**Presentation:** 2
**Contribution:** 2
**Rating:** 4
**Confidence:** 4

**Summary:**

This paper introduces the Athenian Academy, a seven-layer architecture for Multi-Agent Systems (MAS), inspired by the OSI model in computer networking and designed to bring structure, reproducibility, and analytical clarity to LLM-based multi-agent systems. The framework systematically decomposes MAS into seven conceptual and functional layers, progressing from micro-level interactions to macro-level synthesis

**Strengths:**

The seven-layer abstraction is a novel and principled attempt to formalize the design of LLM-based MAS. The analogy to the OSI model is insightful, offering a structured vocabulary for what has so far been a largely ad-hoc field. Layer 7’s explicit integration of social choice theory for ethical arbitration is a creative and meaningful advancement toward embedding values and fairness into system architecture.

**Weaknesses:**

- Experiments are restricted to the artistic creation domain, which, while illustrative, may not fully represent the performance of the architecture in domains with stricter task constraints (e.g., scientific reasoning, robotics, finance). Although Section 4.3 discusses generalizability, actual cross-domain evidence is missing.
- The architecture introduces significant computational overhead through multi-layer orchestration and memory management. There is little quantitative discussion of efficiency trade-offs, e.g., latency, resource usage, or scaling behavior as agent numbers grow.
- While each layer is validated independently, there is no ablation study exploring interactions between layers or potential redundancy (e.g., whether Layers 5 and 6 are complementary or overlapping).

**Questions:**

- How tightly coupled are the layers? Could some be collapsed or skipped for simpler tasks?
- Is there a formal dependency graph among layers that defines necessary or optional relationships?
- Can the authors quantify the computational overhead of maintaining this seven-layer stack compared to flat MAS architectures?
- Are there adaptive mechanisms to prune or dynamically activate layers depending on task complexity?

---

### Official Review · Reviewer_MSsY · 2025-11-01

**Soundness:** 1
**Presentation:** 1
**Contribution:** 2
**Rating:** 0
**Confidence:** 3

**Summary:**

The paper introduces the Athenian Academy, a seven-layer architecture for Multi-Agent Systems (MAS) that organizes system design into structured, analyzable components. It describes each layer as addressing a distinct aspect of multi-agent interaction, providing a taxonomy for classifying and comparing agent capabilities. Experimental evaluations in the context of AI-based artistic creation illustrate how the layered structure affects collaboration, thematic consistency, and knowledge exchange among agents.

**Strengths:**

The paper introduces an ambitious framework that attempts to bring structure and reproducibility to Multi-Agent System (MAS) design.

The analogy to layered architectures provides an intuitive conceptual anchor.

The focus on AI-driven art creation as a test domain is interesting and challenges conventional evaluation settings.

**Weaknesses:**

Arbitrary Layer Design: The choice of seven layers feels arbitrary and lacks clear theoretical or empirical justification.

Overlap Between Layers: Several layers describe similar concepts without clear distinctions or rationale for separation. For example, “Multi-Agent Collaboration” and “Multi-Agent Collaboration via a Shared Large Model"

Unclear Terminology: Terms like “Single-Agent Multi-Role Playing” and “Cross-Scene Experience Traversal” are vague and metaphorical, lacking operational definitions that would make them measurable or technically precise.

Inconsistent Layer Boundaries: The framework mixes conceptual levels; some layers describe mechanisms (“Leveraging Multiple Large Models”), while others describe outcomes or properties (“Task Adaptability”), making the hierarchy logically inconsistent.

Arbitrary Progression: The transitions between layers do not follow a clear developmental logic.

Redundant Focus on Adaptability: Many layers reiterate similar ideas about “adaptability” and “flexibility,” indicating potential redundancy that could be consolidated into fewer, more distinct categories.

Insufficient Empirical Justification: The evaluation of the layers is not clearly explained, and the claimed quantitative experiments are not presented in sufficient detail.

Taxonomy Clarity: The taxonomy appears arbitrary and insufficiently grounded in prior theoretical or empirical frameworks.

Presentation Choices: The inclusion of the painting (Figure 1) and artistic avatars such as “Da Vinci” detract from the professional tone and technical focus of the paper.

Stylistic Tone: Some phrasing, such as “Style Quantum Entanglement,” comes across as non-rigorous for a technical paper.

Conceptual Framing: The framework might be more coherent if presented as a set of functional modules or tasks rather than as rigid hierarchical layers.

Coherence and Structure: The paper lacks a cohesive overarching theme and instead reads like seven loosely connected conceptual scenarios.

Long-Term Relevance: The approach seems at odds with the “bitter lesson,” relying on hand-crafted structure rather than scalable general methods. It is unclear whether the proposed framework will remain relevant or generalizable beyond the current experimental context.

**Questions:**

What is the theoretical or empirical rationale for selecting seven layers, and why is this particular level of granularity necessary?

How could the paper better clarify and differentiate the purpose of each layer to reduce redundancy and overlap?

How might the conceptual hierarchy be reorganized so that all layers represent a consistent level of abstraction?

---

### Meta-Review · Area_Chair_WssL · 2026-01-05

**Summary:**

This paper proposes a novel architecture for MAS systems, consisting of decomposing the architecture into a seven-layer architecture. The seven layers are inspired by an Athenian academy, and the authors provide several creative use cases of the approach. Reviewers were concerned about how ad hoc the system was and that the overall evaluation was done on very limited domains. Since all reviewers recommended rejection and the authors did not submit a rebuttal, the AC recommends rejection.

**Reviewer Concerns:**

Reviewers were concerned that the entire system is rather ad hoc and that the overall evaluation was done on very limited domains.

**Reviewer Scores:**

Since the authors did not submit a rebuttal, I do not think the reviewers would have updated their scores.

---

### Decision · Program_Chairs · 2026-01-26

Reject